# Evaluating the Severity of Autism Spectrum Disorder from EEG: A Multidisciplinary Approach Using Statistical and Artificial Intelligence Frameworks

**DOI:** 10.3390/bioengineering12111225

**Published:** 2025-11-10

**Authors:** Noor Kamal Al-Qazzaz, Sawal Hamid Bin Mohd Ali, Siti Anom Ahmad

**Affiliations:** 1Department of Biomedical Engineering, Al-Khwarizmi College of Engineering, University of Baghdad, Baghdad 47146, Iraq; 2Department of Electrical, Electronic and Systems Engineering, Faculty of Engineering and Built Environment, Universiti Kebangsaan Malaysia, Bangi 43600, Selangor, Malaysia; sawal@ukm.edu.my; 3Centre of Advanced Electronic and Communication Engineering, Department of Electrical, Electronic and Systems Engineering, Universiti Kebangsaan Malaysia, Bangi 43600, Selangor, Malaysia; 4Department of Electrical and Electronic Engineering, Faculty of Engineering, Universiti Putra Malaysia, Serdang 43400, Selangor, Malaysia; sanom@upm.edu.my; 5Malaysian Research Institute of Ageing (MyAgeing) TM, University Putra Malaysia, Serdang 43400, Selangor, Malaysia

**Keywords:** Autism, Wavelet, Statistical analysis, ANOVA, Pearson’s correlation, Machine learning, deep learning, LSTM

## Abstract

A developmental impairment known as autism spectrum disorder (ASD) impacts youngsters and is characterized by impaired social communication and limited behavioral expression. In this study, electroencephalography (EEG) is used to obtain the brain electrical activity of typically developing children and of mild, moderate, and severe ASD patients using relative powers. This study investigates ASD patients using a multidisciplinary approach involving two-way ANOVA and Pearson’s correlation statistical analyses to better understand the multistage severity of ASD from EEG by providing a spectro-spatial profile of ASD severity. Artificial intelligence frameworks, including a decision tree (DT) machine learning classifier and a long short-term memory (LSTM) neural network, are applied to discriminate mild, moderate, and severe ASD patients from typically developing children. The statistical results revealed that with increasing severity compared to the control, faster frequencies decreased and slower frequencies increased, indicating a distinct correlation between the severity of ASD and neurophysiological activity. Moreover, the DT classifier achieved a classification accuracy of 65%, and the LSTM classifier achieved a classification accuracy of 73.3%. This approach highlights the potential for statistical and artificial intelligence techniques to reliably identify EEG abnormalities associated with ASD, which could lead to earlier treatment and improved prospects for patients.

## 1. Introduction

Brain disorders impact a large portion of the world’s population and are caused by a complex interplay of biological, social, and psychological factors. Autism is one of these neurodevelopmental disorders that typically appears in children [1].

According to research conducted by the World Health Organization (WHO), the treatment of mental disorders has not kept pace with the increase in these disorders, which have lasting and detrimental effects on the lives of affected children and their families, as well as on society as a whole [2,3].

The detrimental effects of ASD on families and children include difficulties adjusting to new environments, difficulties in the classroom, increased levels of stress and psychological strain, and, to differing degrees in different cultures, societal and economic setbacks [2]. This is reflected in the rise in costs associated with this group, including social, educational, and healthcare support; costs associated with low financial return as a result of unemployment; and costs associated with caring for people with ASD across all age groups [4].

In order to identify the root causes of autism, medical intervention from experienced physicians is necessary. Observing the behavior of affected children in various settings and the challenges they face when interacting with their surroundings are the main components of an autism diagnosis [4].

Although it shares symptoms with other mental illnesses, ASD is a complicated disorder in and of itself. Therefore, it is crucial to diagnose individuals with ASD using suitable evaluation tools [3]. According to Radhakrishnan [5], there are a variety of tests that may be used to determine whether a child has autism, including behavioral evaluation and occupational therapy screening. The Childhood Autism Rating Scale (CARS), the Autism Diagnostic Observation Schedule (ADOS) [6], and the Autism Diagnostic Interview–Revised (ADI-R) [7] are some of the clinical procedures commonly used to evaluate autism spectrum disorder. But we must not forget that these evaluations are not interchangeable; they each have their own set of strengths and weaknesses. These evaluations are defined by the amount of time they take, the breadth of the questions they cover, and the requirement that licensed professionals must administer them [5,8,9].

Several time–frequency analysis methods can be applied to biomedical images and data in order to identify and diagnose a wide range of human disorders [5,10]. In addition, several anomalies in brain electrical activity can be detected using electroencephalogram (EEG) tests and analyses [11,12,13].

EEG can be utilized to assist with the early detection, management, and prevention of autism by revealing complex relationships. Early in the first year after birth, EEG waveforms, in particular the gamma (γ) band from the frontal brain lobe, can differentiate ASD from normal controls [14]. In addition, statistical learning approaches can use the non-linear properties of EEG to differentiate between individuals with ASD and those without [3,15].

Indeed, there have been tremendous improvements in the EEG areas of diagnosis, evaluation, and treatment of people with ASD, particularly due to the use of artificial intelligence in autism research, detection, and classification [16]. As an essential part of artificial intelligence, machine learning uses different algorithms like support vector machine (SVM), K-nearest neighbors (KNN), and decision tree (DT) [5,17].

A subfield of machine learning is deep learning, which uses artificial neural networks designed to mimic the structure of the human brain. Deep learning is highly effective in image identification, speech analysis, and natural language processing, all of which impact our ability to understand and treat autism [5,17]. Neural networks, including recurrent neural networks (RNNs), particularly long short-term memory (LSTM) networks, have been widely used to analyze time-series data. Convolutional neural networks (CNNs) have been utilized to analyze neuroimaging data and video and have improved diagnostic accuracy while also reducing the influence of subjectivity [14,18,19,20].

Neurobiological studies have recently attracted much attention. Thus, this study integrates statistical analyses with machine learning and deep learning algorithms to capture the heterogeneity of autism. The EEG dataset contains recordings from mild ASD patients, moderate ASD patients, severe ASD patients, and typically developing children. Conventional filters and discrete wavelet transforms (DWTs) are used in the preprocessing stage to denoise the EEG data. Relative power frequency-domain features are extracted to obtain the spectro-spatial EEG profile. The first part of this study looks at relative power features to create spectral neuro-markers, using two-way analysis of variance (ANOVA) and Pearson’s correlations to identify spectro-spatial ASD profiles in various brain regions. A comparative study is then conducted to autonomously determine ASD severity from EEG signals using DT machine learning-based classification algorithms and LSTM deep learning-based classification models.

The objectives of this study are to analyze the relative power features using statistical analyses to characterize the spectro-spatial profile of ASD severity using EEG data and to assess the effectiveness of machine learning and deep learning models in determining ASD severity. Thus, it aims to discriminate autism through the characterization of temporal changes in physiological EEG activity in these individuals by identifying EEG alterations in children with ASD and assessing aspects of their responses in relation to the severity of ASD, according to a conventional statistical method of ANOVA that is employed to compare differences in ASD severity through relative power spectrum coefficients between children with ASD and neurotypicals. Furthermore, Pearson’s correlation approach is employed to define the correlation that exists between adaptive behaviors and autism severity in order to determine the dimensions of these aspects. Furthermore, a DT and an LSTM network are employed for time-series modeling, separating sequences of brain activity associated with diagnosing and treating autism.

The combination of statistical analyses with machine and deep learning techniques improves not only the quality of the outcomes derived from studies on autism but also the prospect of discovering the spectro-spatial profile of ASD severity.

This study makes three contributions: it is the first to use a combination of relative power features to develop a spectro-spatial profile identification for ASD severity in brain regions using statistical analyses; second, it is the first study that automatically manages ASD severity as a multiclassification problem of autism severity, specifically separating mild, moderate, and severe autistic symptoms from typically developing control subjects; and third, the ASD EEG-based dataset has never been used before.

## 2. Related Works

Clinical diagnostic criteria rely on behavioral observations and therefore often result in delayed therapeutic intervention and less-than-favorable outcomes for patients with ASD [21,22]. Recently, interest has shifted toward using neuroimaging techniques, like MRI and fMRI, and neurophysiological tools, including EEG, to objectively define ASD [23,24].

EEG has several important advantages, such as its non-invasive nature, temporal resolution, and ability to record dynamic neural activity that is fundamentally related to deficits observed in individuals with ASD [25,26]. The related literature has examined the use of EEG features like power spectral densities, coherencies, and complexity measures to distinguish between ASD patients and healthy controls [27,28,29,30].

In recent years, ASD classification has been investigated using features extracted from EEG signals with the aid of machine learning algorithms. Features derived from raw data can be used to train a machine learning classifier to diagnose ASD [31].

However, the majority of existing studies on EEG-based ASD diagnosis relied solely on binary classification (ASD vs. control) or did not consider the severity of symptoms at all [32,33]. More integrative methods need to be developed to accurately quantify the pathology of ASD from EEG signals, monitor the progression of the disease, and provide more targeted intervention and treatment plans [34,35].

Brihadiswaran [17] discussed EEG-based ASD classification using machine learning approaches and highlighted the need for early identification and a standardized method of diagnosing the disorder. Moreover, Heunis et al. [36] applied recurrence quantification analysis (RQA) with linear discriminant analysis (LDA) and support vector machine (SVM) classifiers. Haputhanthri et al. [37,38] applied statistical features with SVM, naive Bayes (NB), and random forest (RF) and emphasized the potential of more complex mathematical techniques through the investigation of Shannon entropy with the linear regression (LR) classifier. However, researchers have also used machine learning to analyze genetic data and identify biomarkers associated with autism. This has helped us learn more about the genetic profile of the disorder [39].

Deep learning’s rapid growth demonstrates its ability to extract features, thereby increasing the accuracy of classification, reducing human intervention and data processing times, and helping doctors in this field with diagnoses and filling gaps in traditional approaches. Advanced neural networks are now being used to analyze EEG datasets related to autism, particularly convolutional neural networks (CNNs), which can identify interaction patterns in large-scale neuroimaging data [40].

Thus, new research using deep learning could help us determine how EEG data recorded from people with neurological disorders such as autism can be more effectively utilized. Based on exploratory research conducted over the past few years, convolutional neural networks (CNNs) and recurrent neural networks (RNNs) have shown that EEG data can be used to detect neurological disorders. They are capable of learning from data and sometimes outperform traditional machine learning methods [41,42].

Indeed, deep learning enables the extraction of additional valuable information from EEG signals. For instance, Hasan et al. [43] applied CNN, DNN, and LSTM models to demonstrate their effectiveness in handling complex data like EEG signals. Also, Al-Qazzaz et al. [18] used EEG to determine the severity of ASD in patients after encoding EEG signals into images to make them easier to interpret by deep learning models. They showed that using both EEG features and images improved classification accuracy when using a pre-trained SqueezeNet CNN model [18].

Several studies have used EEG-based datasets to address ASD-related challenges. Table 1 presents a summary of research on distinguishing between ASD and normal controls using EEG analysis with the aid of machine and deep learning techniques such as SVM, NB, RF, LR, and DNN [5,37,38,44,45].

For machine learning, feature extraction techniques have included statistical measures, power spectral densities, and entropy [5,37,38,44,45]. In these studies, classification accuracy ranged from 56% to 94%. However, most of these studies relied on machine learning classifiers, indicating the ability of such methods to distinguish specific patterns between ASD individuals and those who are typically developing. Deep learning-based approaches have also achieved high classification accuracy, suggesting that deep learning could enhance diagnostic tools and guide future therapeutic interventions. While previous studies have utilized deep learning to distinguish between typically developing children and autistic individuals, what sets our study apart is its focus on the multiclassification problem of autism severity, specifically distinguishing mild, moderate, and severe autistic symptoms from typically developing controls.

**Table 1 bioengineering-12-01225-t001:** Related works.

Studies (Year)	Extracted Features	Classification Approaches	Classification Accuracy (%)
Haputhanthri et al. [37]	Statistical features (mean and standard deviation)	SVM, NB, RF	53.33, 73.33, 66.66
Haputhanthri et al. [38]	Shannon entropy, statistical methods	LR	94
Radhakrishnan et al. [5]	Automatic feature extraction and classification	DNN	81
Garcés et al. [44]	Power spectrum	ElasticNet	56, 64
Hou et al. [45]	*t*-test, PCA, ReliefF, Chi-square	SVM	74.1

In general, the studies indicate a progression from empirical-type methodologies to machine learning-based methodologies that could be useful in the diagnosis of ASD, but there is still a lot to be done in this area in terms of having a standardized method and increasing the accuracy of diagnostic prediction across different datasets.

This research seeks to fill this gap by adopting an interdisciplinary approach that uses statistical methods and artificial intelligence, including machine learning and deep learning models, to assess ASD severity from EEG signals. Hence, this work provides an objective and more direct estimation of ASD severity, which could potentially be helpful for patients in terms of diagnosis and treatment.

## 3. Materials and Methods

Preprocessing, feature extraction, and classification are all necessary steps in processing and analyzing recorded EEG signals in order to evaluate ASD severity using EEG signals. Figure 1 is a schematic depicting the proposed strategy.

The implementation was carried out on a computer with a 13th Gen Intel Core i7-13650HX 2.6 GHz CPU, which offers high processing power appropriate for model training and numerical calculations. To expedite the LSTM model’s training, we employed an NVIDIA GeForce RTX 4060 GPU. This GPU made it possible to handle large-scale EEG datasets efficiently by drastically cutting down on training time. During model training and validation, the system’s 16 GB of RAM allowed for seamless data handling and processing. In order to ensure compatibility with MATLAB R2024b and its related toolboxes, the implementation was conducted in a Windows 11 environment.

### 3.1. ASD EEG-Based Dataset

The EEG dataset consists of recordings from forty subjects, including ten normal subjects, comprising five females and five males, with an average age of 8.545 ± 1.1 years; ten mild autism spectrum disorder patients, comprising four females and six males. In the case of moderate ASD, 10 cases, where the average age of the cohort is 8.364 yrs old with a standard deviation of 0.8 yrs, 3 of the ten are female, and 7 are male, with a mean age of 8.727 yrs old and standard deviation of 0.98 yrs old; and 10 cases of severe autism spectrum disorder, where the average age of the cohort is 8.727 yrs old with a standard deviation of 0.98 yrs.

The patients were the participants in the research conducted both at the Neurophysiology Department of the Baghdad Teaching Hospital and the Autism Centre of the Paediatric Hospital in Medical City, Baghdad, Iraq. No children included in the ASD had taken medication within the two weeks before the recording of the EEG. The children of the control group did not have any family record of a neurological or mental disorder and were enrolled in normal, health promoting schools.

The interviews of the children were done by a psychiatrist based on the provisions set in *Diagnostic and Statistical Manual of Mental Disorders* (DSM-V) [46]. Moreover, subjects were assessed for the severity of ASD based on the Gilliam Autism Rating Scale (GARS-3) [47].

**Figure 1 bioengineering-12-01225-f001:**
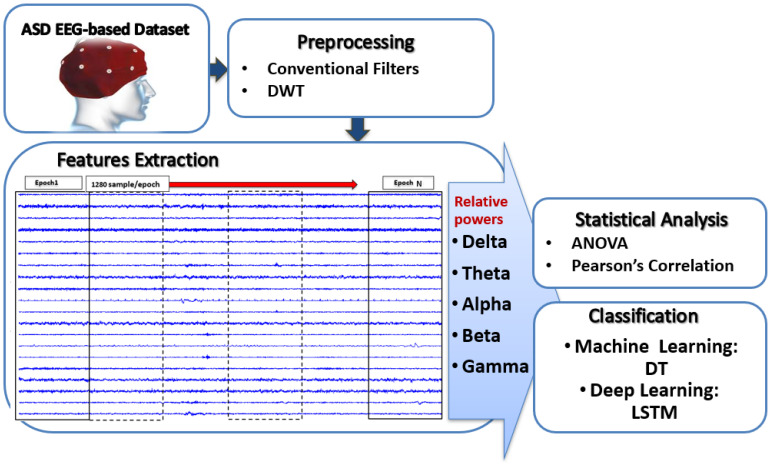
Schematic diagram of assessing ASD severity from EEG signals.

A device manufactured in Japan by Nihon Kohden was required to record a 10 min EEG signal. Nineteen Ag/AgCl electrodes were laid as per the 10–20 system and two reference electrodes were placed on the mastoid bones. The data was collected with an impedance of less than 5 K if the impedance, a resolution of 12 bits and a sampling rate of fs=256 Hz. When the recording was recorded, the bandpass filter was adjusted between 0.1 and 70 Hz and the notch filter was adjusted to 50 Hz. The data was filtered to be analyzed in the range of bandpass filters of 0.1 to 64 Hz.

This study followed the 1964 Helsinki Declaration with all its amendments, and the principles of the medical school and the institutional research committee, which was held in the College of Medicine of the University of Baghdad. The consent of the parents of the children used in the study was obtained after all the necessary information was received.

### 3.2. Preprocessing

In the initial stage of processing, conventional filters, including a notch filter at 50 Hz, were employed to remove interfering noise, and a bandpass filter with a frequency range of 0.1–64 Hz was used to limit the band of the recorded EEG signals [48,49].

After that, each of the 19 denoised EEG channels had its signal split into non-overlapping epochs of 5 s length, and each epoch contained N=1280 samples, as the sampling frequency was set at fs=256 Hz. Segmenting the EEG channels significantly increased the amount of data available for analysis, effectively enhancing the dataset’s dimensionality.

The 5-second segmentation was preliminary and supported by a trade-off between temporal precision and EEG data analysis. EEG signals exhibit large frequency changes; therefore, 5-second is enough to capture brain electrical activity from delta to (gamma(γ), beta(β), alpha(α), and theta(θ)) bands. Each 5-second segment was expected to capture EEG signal transients and structures. This study used a length of time that has been employed in other EEG analysis-based studies to discriminate neurophysiological properties for distinct states or disorders, including ASD [18,50,51]. To maintain neurophysiological features for meaningful segmental assessment, all channels were filtered before obtaining the segments. In the next step, all the attributes were extracted from each 5-second sequence and related to the 10-min dataset.

As a denoising approach, wavelet transforms are widely used. Recently, they have been utilized to process non-stationary data such as EEG signals. Zikov et al. and Krishnaveni et al. utilized wavelet transforms to eliminate ocular artifacts [52,53], and other studies have demonstrated that discrete wavelet transforms (DWTs) are effective for denoising and feature extraction. Therefore, DWTs were used in this study to denoise the raw EEG data. Due to their time-invariant qualities and enhanced time resolution, the DWT is a powerful method for recognizing patterns, detecting changes, and extracting features through EEG wave decomposition [54,55].

Because of their impact on denoising and decomposition, choosing the mother wavelet and decomposition level is crucial. The mother wavelet is ideal for identifying changes in EEG signals and offers superior accuracy compared to other wavelets. Furthermore, its sinusoidal shape and increased stretch on the time axis make it an attractive choice for EEG denoising [56,57].

The coiflet mother wavelet order 3 (coif3) was used in this study, as it is perfect for spiky artifact removal, which includes eye movements/blinks and muscular movements [58]. Moreover, the universal threshold was used as the main thresholding method in this study [56,57].

### 3.3. Feature Extraction

The time-invariance qualities of the signal are crucial, and the DWT can preserve these, particularly when locating and recognizing changes or transient features in the EEG [54,59,60].

These benefits are a result of the DWT method’s multi-resolution analysis characteristic, which is why it is widely used by researchers to analyze EEG data [58]. Therefore, the DWT technique was employed to denoise and decompose the EEG signals in this study.

For this reason, the impulse responses of the EEG signals were separated into low-pass and high-pass filters according to the selected wavelet criteria. Coefficients of approximation (A), containing low-frequency components, are produced by low-pass filtering. The detail (D) coefficients, containing high-frequency components, are produced by high-pass filtering. Consequently, the DWT is a suitable technique for acquiring brain rhythms with various frequency components [56,57].

Each filter halved the signal frequency because the input EEG signal was 256 Hz. As a result, 128 Hz was the frequency of the low-pass and high-pass filters. Following the filtering phase, the EEG signals were divided into their components using the DWT technique and the coif3 mother wavelet.

Since the EEG input used in this experiment had a sample rate of 256 Hz, five decomposition levels were necessary to extract the desired brain rhythms. The decomposition (D1) occurred at a frequency of 64–128 Hz, which was considered noise; however, D2, D3, D4, and D5 represented the (γ, β, α, and θ) rhythms at 30 to 64, 16 to 30 Hz, 8 to 16 Hz, and 4 to 8 Hz, respectively. The approximation (A5) also represented the delta(δ) wave at 0 to 4 Hz.

Previous studies have revealed changes in the (γ, β, α, and θ) bands of neural oscillations, leading researchers to posit that EEG-based markers may help improve the diagnosis and assessment of ASD.

### 3.4. Statistical Analysis

To begin with, the first step involved categorizing the 19 channels of the EEG data of healthy people and those with different severity of ASD into 5 recording regions that represented different parts of the cerebral cortex in the scalp. These were the channels of the frontal, temporal, parietal, central and occipital respectively. Normality was then tested using KolmogorovSmirnoff. There were two statistical analysis sessions carried out in SPSS 23.

#### 3.4.1. ANOVA

The first statistical analysis conducted was a two-way ANOVA. The dependent variable was the relative power in (δ, θ, α, β, and γ), while the independent variables were the group factors (healthy age-matched typically developing children, mild ASD, moderate ASD, and severe ASD) and the brain lobes (frontal, temporal, parietal, and occipital). Post hoc comparisons were made using Duncan’s test. A significance level of p<0.05 was established for all statistical tests.

#### 3.4.2. Pearson’s Correlation

Additionally, in order to assess the correlation of the proposed features, Pearson’s correlation coefficient (*r*) was utilized to calculate correlations between the high-frequency relative powers (α, β, and γ) and the low-frequency band relative powers (δ and θ). Patients with mild, moderate, and severe ASD were the subjects of the Pearson’s correlation session. Two levels of significance were established for Pearson’s correlations: p<0.05 and p<0.01.

### 3.5. Classification

To distinguish between individuals with ASD and typically developing children, machine learning and deep learning classification models were used. Initially, the frequency-domain relative power features were extracted from the EEG signals as part of this research. Then, the DT machine learning classifier and the LSTM RNN deep learning model were applied to obtain the classification results.

#### 3.5.1. Machine Learning Classification

The DT served as the final stage for distinguishing between individuals with ASD and typically developing children. Utilizing decision trees enables a high degree of interpretability through their decision structure, and they handle numerical and categorical inputs well, making them suitable for this task. Decision trees provide great benefits in clinical environments since they offer reasonable model decisions combined with simple communication for practitioners. Field-based research on ASD classification has previously used decision trees [61,62,63,64].

The DT is a recursively generated tree that uses the supervised classification technique to divide a large problem into smaller subproblems. In a DT, each leaf node is assigned a class label, while internal nodes and the parent node utilize attribute testing criteria to divide records according to their features [65]. The number of trees does not significantly impact classification accuracy. In this investigation, an ensemble of 100 binary decision trees was used.

#### 3.5.2. Deep Learning Classification

In this study, the RNN deep learning classification model was utilized to distinguish between individuals with ASD and typically developing children. LSTM effectively identifies temporal connections within sequential data, such as EEG signals, thereby enabling appropriate analysis for time-series data. LSTM networks were used since they have a remarkable ability to identify long-term dependencies and, when combined with EEG data oscillations, can indicate ASD severity. Several studies have used LSTM networks for neurophysiological data classification, supporting our choice [66,67,68,69].

RNNs suffer from the vanishing gradient problem, making them difficult to use for practical purposes. However, LSTM helps reduce the multiplication of gradients smaller than zero [70]. An LSTM consists of the following components:Input gate: uses the tanh activation function, as given by Equation (Equation 1):(1)$$k=tanhb\left(t\right)+x_{t}W_{1}^{\left(k\right)}+y_{t-1}W_{2}^{\left(k\right)}$$
where the current input is $x_{t}$ and the previous cell’s output is $y_{t-1}$.The input gate, a sigmoid-activated node in the hidden layer, is expressed in Equation (Equation 2):(2)$$I=\sigma b^{\left(t\right)}+x_{t}W_{t}^{\left(t\right)}+y_{t-1}W_{2}^{l}$$The output of the input gate is(3)k∘IForget gate: delays the inner state of the LSTM by one time step and adds it to Equation (Equation 3). This creates an internal recurrence loop that learns the relationship between the inputs given at various times. This stage includes sigmoid-activated nodes that decide which previous state should be remembered. The forget gate is expressed in Equation (Equation 4):(4)$$F=\sigma \left(b^{\left(F\right)}\right)+x_{t}W_{1}^{\left(F\right)}+y_{t-1}W_{2}^{\left(F\right)}$$The output of this stage is obtained using Equation (Equation 5):(5)$$o_{t}=o_{t-1}\circ F+k\circ I$$
where $o_{t-1}$ represents the inner state of the previous cell.Output gate: is composed of a tanh squashing function and an output sigmoid function, as expressed in Equation (Equation 6):(6)$$L=\sigma (b^{\left(L\right)}+x_{t}W_{1}^{\left(L\right)}+y_{t-1}W_{2}^{\left(L\right)})$$The cell’s output is calculated as(7)$$y_{t}=tanh\left(o_{t}\right)\circ L$$
where $b^{\left(i\right)}$, $W_{1}^{\left(i\right)}$, and $W_{2}^{\left(i\right)}$ represent the input bias, input weight, and preceding cell’s output for each stage. The weights and biases are set during the training phase.

Ten mild, ten moderate, and ten severe ASD patients, as well as ten typically developing children, were classified using the LSTM RNN model with an initial input size of 5 features×19 channels per feature. To classify patients as normal, mild ASD, moderate ASD, or severe ASD, the LSTM RNN contained 100 hidden units and the following layers: a feature input layer, a fully connected layer, the rectified linear unit (ReLU) activation function, a softmax layer, and lastly, a classification output layer. Using a learning rate of 0.001 and a batch size of 64, the RNN was trained using the adaptive moment estimation (Adam) optimizer across 30 epochs.

### 3.6. Model Evaluation Metrics

In this research, classification accuracy was calculated by first using the DT machine learning classification model and then the LSTM RNN to classify the EEG dataset into four classes: mild ASD, moderate ASD, severe ASD, and normal controls. To prevent class imbalance, the dataset was split into 80% for training and validation and 20% for testing (80:20).

The scope of the evaluation was expanded to include segment- and event-based performance results. To assess ASD prediction performance, the true positive (TP) rate was defined as the number of EEG segments correctly classified as belonging to a specific ASD category, the true negative (TN) rate as the number of segments correctly classified as not belonging to that category, the false positive (FP) rate as the number of segments incorrectly classified as belonging to the category, and the false negative (FN) rate as the number of segments incorrectly classified as not belonging to it. These parameters were used to calculate accuracy, sensitivity, specificity, precision, and F1-score.

## 4. Results and Discussion

### 4.1. Preprocessing Results

For each participant, a 10-min (153,600 sample) EEG recording was preprocessed using the DWT technique with the coif3 denoising mother wavelet and then segmented into 120 non-overlapping 5-second epochs, with dimensions of 120 epochs×19 channels. The resulting dataset was then analyzed for mild ASD, moderate ASD, severe ASD, and normal activity.

### 4.2. Results of Statistical Analysis

Characterization was performed on a total of 153,600 subjects across 19 EEG channels over 10-min intervals. The EEG recordings were divided into 5-second segments, with 1280 data points per segment, for subjects with mild ASD, moderate ASD, severe ASD, and those typically developing. The results from the feature extraction process were analyzed using ANOVA and Pearson’s correlation, two approaches often used in statistical analysis.

#### 4.2.1. ANOVA Results

It was observed that different parts of the brain exhibited markedly different patterns of activity in relation to each frequency band. With the greatest relative strength in the delta band under the severe condition, the five brain regions displayed clear variations in the α, β, γ, and θ bands across ASD severities Figure 2. This may indicate that increasing severity has a greater impact on brain activity in the slower delta frequency band, whereas typical conditions have a greater impact on the θ, α, and β bands.

δ band power appeared to rise incrementally with the severity of the disorder across all regions. α band power, meanwhile, was more conditionally stable. The α frequency may be intrinsically stable and may not reflect the severity of the disorder to the same extent as other regions or frequencies in the frontal, parietal, temporal, and occipital areas.

A remarkable difference was also observed in the θ bands across the five regions. These bands had high relative power under typical conditions, which could suggest a link between reduced θ slow-wave activity and moderate ASD.

In the frontal, parietal, occipital, and temporal lobes, there was a distinct pattern in which the relative strengths of α and β appeared to remain fairly consistent regardless of severity. This might mean that α and β activity in these regions respond to variations in severity by reducing their power relative to typical conditions.

In conclusion, the moderate condition was characterized by a large increase in the relative power of the γ band in the central, parietal, and occipital regions, in contrast to other bands that did not exhibit a similarly notable rise. This indicates that increasing severity relative to typical conditions caused a decrease in faster frequencies such as α, β, and γ, and an increase in slower frequencies such as δ and θ.

Bonferroni adjustments were applied to each relative power in multiple comparisons of the brain lobes, as shown in Table 2, Table 3, Table 4, Table 5 and Table 6.

**Figure 2 bioengineering-12-01225-f002:**
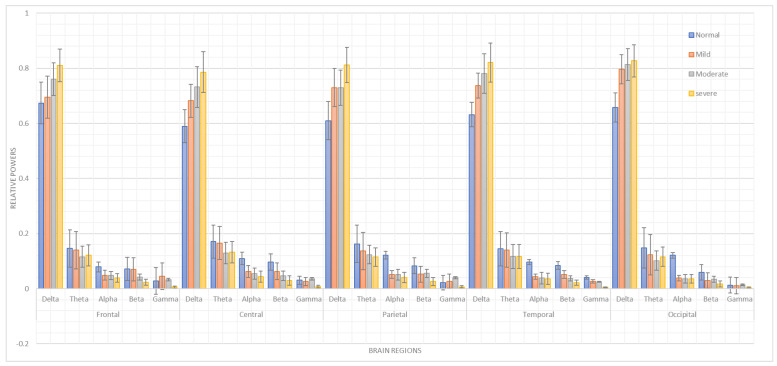
Comparison the relative power features across five scalp regions of the brain among patients with mild, moderate, and severe ASD, and typically developing children.

**Table 2 bioengineering-12-01225-t002:** Frontal lobe multiple comparison test using the Bonferroni correction for relative power as the dependent variable.

Powers	(I) ASD Class	(J) ASD Class	Sig.
δ	Moderate	Mild	0.045
		Severe	0.125
		Normal	0
	Mild	Severe	0
		Normal	0.817
	Severe	Normal	0
θ	Moderate	Mild	0.152
		Severe	0.951
		Normal	0.005
	Mild	Severe	0.358
		Normal	0.928
	Severe	Normal	0.033
α	Moderate	Mild	0.999
		Severe	0.614
		Normal	0
	Mild	Severe	0.625
		Normal	0.001
	Severe	Normal	0
β	Moderate	Mild	0.011
		Severe	0.125
		Normal	0
	Mild	Severe	0
		Normal	1
	Severe	Normal	0
γ	Moderate	Mild	0.497
		Severe	0.003
		Normal	0.866
	Mild	Severe	0
		Normal	0.124
	Severe	Normal	0.009

#### 4.2.2. Pearson’s Correlation Results

Pearson’s correlation was employed to examine the relationships between the lower-frequency bands (δ and θ) and the higher-frequency bands (α, β, and γ) across various regions of the brain and among patient groups (mild, moderate, severe, and normal), as shown in Table 7. This analysis revealed substantial correlations that provide insights into the underlying neural processes.

In the frontal region, Pearson’s correlation coefficients indicated strong negative correlations between δ power and the higher-frequency bands: α (r = −0.723, p<0.05), β (r = −0.881, p<0.05), and γ (r = −0.638, p<0.05). These results imply that as δ power increased, the α, β, and γ powers decreased. In contrast, θ activity did not show a consistent relationship with the higher-frequency bands. Overall, the observed pattern supports a negative correlation between δ and higher-frequency activity, and a weaker or variable correlation between θ and these frequencies.

**Table 3 bioengineering-12-01225-t003:** Central lobe multiple comparison test using the Bonferroni correction for the relative power as the dependent variable.

Powers	(I) ASD Class	(J) ASD Class	Sig.
δ	Moderate	Mild	0.781
		Severe	0.671
		Normal	0.007
	Mild	Severe	0.213
		Normal	0.238
	Severe	Normal	0
θ	Moderate	Mild	0.228
		Severe	0.998
		Normal	0.038
	Mild	Severe	0.299
		Normal	0.991
	Severe	Normal	0.062
α	Moderate	Mild	0.984
		Severe	0.919
		Normal	0.004
	Mild	Moderate	0.984
		Severe	0.788
	Severe	Normal	0
β	Moderate	Mild	0.833
		Severe	0.743
		Normal	0.008
	Mild	Moderate	0.833
		Severe	0.305
	Severe	Normal	0.208
γ	Moderate	Mild	0.84
		Severe	0.053
		Normal	0.95
	Mild	Moderate	0.84
		Severe	0.445
	Severe	Normal	0.972

Similarly, in the central region, δ power showed negative correlations with α (r = −0.852), β (r = −0.874), and γ (r = −0.661), all significant at p<0.05. Moreover, θ power had a positive correlation with α (r = 0.565, p<0.05) but a weak correlation with β.

In the parietal region, δ power was inversely related to α (r = −0.833), β (r = −0.824), and γ (r = −0.551), all significant at p<0.05. In contrast, θ power showed a significant positive correlation with α (r = 0.446, p<0.05). These results indicate that the parietal region exhibited a pattern similar to that observed in the frontal and central regions: δ power was inversely related to higher frequencies, while θ power had a direct relationship with α and β frequencies. Overall, δ power correlated negatively with α, β, and γ (r = −0.775, −0.900, −0.694, respectively; p<0.05), whereas θ power correlated positively with α and β (r = 0.384, 0.325, respectively; p<0.05). The correlation with γ was insignificant (r = 0.174, p>0.05). This pattern was consistent across regions, demonstrating that δ power was negatively associated with higher frequencies, while θ power was positively associated with α and β activity.

**Table 4 bioengineering-12-01225-t004:** Parietal lobe multiple comparison test using the Bonferroni correction for the relative power as the dependent variable.

Powers	(I) ASD Class	(J) ASD Class	Sig.
δ	Moderate	Mild	1
		Severe	0.336
		Normal	0.04
	Mild	Severe	0.446
		Normal	0.087
	Severe	Normal	0
θ	Moderate	Mild	0.918
		Severe	0.964
		Normal	0.104
	Mild	Severe	0.71
		Normal	0.554
	Severe	Normal	0.029
α	Moderate	Mild	1
		Severe	0.969
		Normal	0.005
	Mild	Moderate	0.972
		Severe	0.019
	Severe	Normal	0.001
β	Moderate	Mild	0.999
		Severe	0.268
		Normal	0.231
	Mild	Moderate	0.432
		Severe	0.281
	Severe	Normal	0.001
γ	Moderate	Mild	0.718
		Severe	0.015
		Normal	0.267
	Mild	Moderate	0.334
		Severe	0.97
	Severe	Normal	0.382

In the occipital region, δ power was negatively correlated with α (r = −0.931), β (r = −0.866), and γ (r = −0.540), with p<0.05 for all correlations. In contrast, θ power was positively correlated with α (r = 0.561, p<0.05) and β (r = 0.436, p<0.05). These results indicate a strong negative correlation between δ activity and the higher frequencies, and a strong to moderate positive correlation between θ activity and α and β frequencies in the occipital region.

### 4.3. Classification Results

In this study, two distinct approaches, DT and LSTM classifiers, were employed to classify ASD severity as mild, moderate, or severe and to distinguish these groups from typically developing children. The performance of the two methods was assessed using precision, sensitivity, specificity, accuracy, and F1-score.

#### 4.3.1. Results of Machine Learning Classification Model

The accuracy of the DT classifier was 65% in classifying the severity of ASD (mild, moderate, or severe) and distinguishing ASD patients from typically developing children. Although this reflects moderate accuracy, it suggests that the model can be improved, especially in differentiating between ASD and control groups.

**Table 5 bioengineering-12-01225-t005:** Occipital lobe multiple comparison test using the Bonferroni correction for the relative power as the dependent variable.

Powers	(I) ASD Class	(J) ASD Class	Sig.
δ	Moderate	Mild	0.995
		Severe	0.995
		Normal	0.026
	Mild	Severe	0.968
		Normal	0.117
	Severe	Normal	0.013
θ	Moderate	Mild	0.85
		Severe	0.937
		Normal	0.119
	Mild	Severe	0.992
		Normal	0.691
	Severe	Normal	0.377
α	Moderate	Mild	1
		Severe	1
		Normal	0.012
	Mild	Moderate	1
		Severe	0.046
	Severe	Normal	0.013
β	Moderate	Mild	0.992
		Severe	0.666
		Normal	0.259
	Mild	Moderate	0.881
		Severe	0.229
	Severe	Normal	0.015
γ	Moderate	Mild	0.983
		Severe	0.283
		Normal	1
	Mild	Moderate	0.597
		Severe	0.987
	Severe	Normal	0.222

Precision varied with the degree of severity, with the highest precision observed for moderate cases (70.7%) and the lowest for severe cases (64.7%). This indicates that the DT model is more reliable in correctly identifying moderate ASD cases compared to normal cases.

Sensitivity showed significant variation according to the severity of ASD. The DT model achieved a sensitivity of 44% for normal cases, increasing to 82% for moderate cases. This suggests that the DT model is more adept at detecting moderate ASD patients.

Specificity remained fairly constant across all severity levels, hovering around 88%. This suggests that the DT model is equally effective at correctly identifying non-ASD cases at each severity level.

When precision and sensitivity were balanced, the F1-score was highest for moderate cases (75.9%) and lowest for normal cases (49.4%). This again highlights the ability of the model to identify cases of moderate ASD.

**Table 6 bioengineering-12-01225-t006:** Temporal lobe multiple comparison test using the Bonferroni correction for the relative power as the dependent variable.

Powers	(I) ASD Class	(J) ASD Class	Sig.
δ	Moderate	Mild	0.744
		Severe	0.736
		Normal	0
	Mild	Severe	0.224
		Normal	0.047
	Severe	Normal	0
θ	Moderate	Mild	1
		Severe	0.115
		Normal	0.451
	Mild	Severe	0.425
		Normal	0.985
	Severe	Normal	0.102
α	Moderate	Mild	0.99
		Severe	0.998
		Normal	0.001
	Mild	Moderate	0.966
		Severe	0.009
	Severe	Normal	0
β	Moderate	Mild	0.72
		Severe	0.529
		Normal	0
	Mild	Moderate	0.113
		Severe	0.037
	Severe	Normal	0
γ	Moderate	Mild	0.999
		Severe	0.441
		Normal	0.546
	Mild	Moderate	0.463
		Severe	0.734
	Severe	Normal	0.017

Figure 3 illustrates the classification performance of the DT model. For the mild class, 34 cases were correctly classified, while 6 were misclassified as moderate, 6 as normal, and 4 as severe. For the moderate class, 41 cases were correctly identified, while 3 were misclassified as severe, 2 as normal, and 6 as mild, indicating some confusion with both mild and severe cases. For the severe class, 33 cases were correctly diagnosed, while 9 were misclassified as normal, 4 as mild, and 4 as moderate, suggesting difficulty distinguishing severe from moderate cases. Lastly, for the normal class, 22 cases were correctly classified, while 8 were misclassified as mild, 7 as moderate, and 13 as severe, reflecting some overlap with the other classes.

**Table 7 bioengineering-12-01225-t007:** Results of Pearson’s correlation between (α, β, and γ) and (δ and θ). *,** Asterisks indicate statistically significant correlations at two-tailed significance levels of 0.05 and 0.01, respectively.

Brain Region	RP	Correlation	α	β	γ
Frontal	δ	*r*	−0.723 **	−0.881 **	−0.638 **
		*p* value	0.05	0.05	0.05
	θ	*r*	0.347 **	0.177 *	−0.055
		*p* value	0.05	0.017	0.462
Central	δ	*r*	−0.852 **	−0.874 **	−0.661 **
		*p* value	0.05	0.05	0.05
	θ	*r*	0.565 **	0.351 **	0.172
		*p* value	0.05	0.002	0.132
Parietal	δ	*r*	−0.833 **	−0.824 **	−0.551 **
		*p* value	0.05	0.05	0.05
	θ	*r*	0.446 **	0.259 *	0.088
		*p* value	0.05	0.022	0.443
Temporal	δ	*r*	−0.775 **	−0.900 **	−0.694 **
		*p* value	0.05	0.05	0.05
	θ	*r*	0.384 **	0.325 **	0.174
		*p* value	0.05	0.001	0.078
Occipital	δ	*r*	−0.931 **	−0.866 **	−0.540 **
		*p* value	0.05	0.05	0.05
	θ	*r*	0.561 **	0.436 **	0.238
		*p* value	0.05	0.001	0.09

#### 4.3.2. Results of Deep Learning Classification Model

The LSTM model exhibited higher accuracy across all severity levels compared to the DT model, with the highest precision for moderate cases (90%) and the lowest for severe cases (65%). This indicates more reliable performance in correctly identifying moderate ASD cases (Table 8).

The sensitivity of the LSTM model was remarkably high in mild cases (100%) but low in moderate cases (60%). This suggests that although the LSTM model is good at detecting mild ASD, it may miss some moderate cases.

Specificity was highest for moderate cases (97.8%) and lowest for severe cases (84.4%). This suggests that the LSTM model was most effective at correctly identifying normal (non-ASD) subjects when distinguishing them from moderate ASD cases.

The overall accuracy of the LSTM model was consistently 73.3%, which was higher than that of the DT model. This suggests that the LSTM model is generally more effective at classifying ASD severity.

**Figure 3 bioengineering-12-01225-f003:**
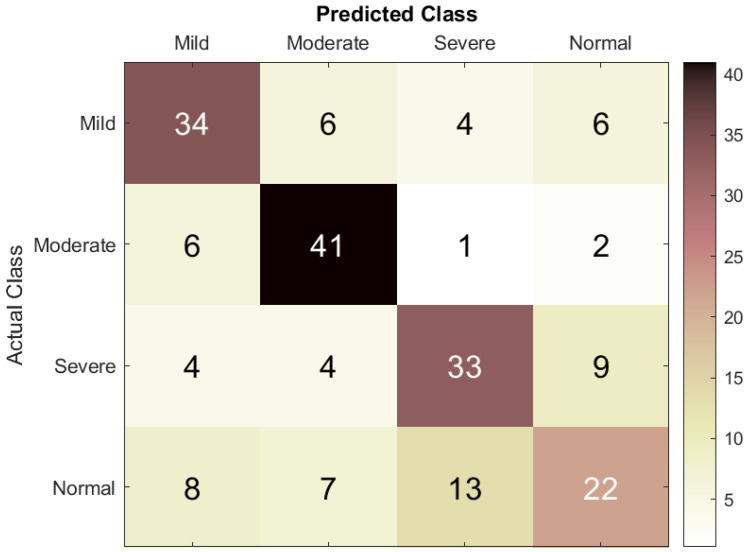
Confusion matrix of the DT model used to assess ASD severity from EEG signals.

The F1-score was highest for mild cases (90.9%) and lowest for moderate cases (72%). This indicates a balanced performance in identifying mild ASD, but performance for moderate cases needs improvement (Table 9).

Comparing the two models, it is clear that the LSTM model outperformed the DT model in most metrics, especially in accuracy, precision, and F1-score. However, the DT model still offers advantages, especially in terms of its specificity and consistent sensitivity to important information. This suggests that a hybrid approach that leverages the strengths of both models could achieve even better classification performance.

**Table 9 bioengineering-12-01225-t009:** Classification results using the relative power features and LSTM model for children with mild, moderate, and severe ASD, and typically developing children.

LSTM	Normal	Mild	Moderate	Severe
Precision (%)	58.3	83.3	90	65
Sensitivity (%)	46.7	100	60	86.7
Specificity (%)	88.9	93.3	97.8	84.4
Accuracy (%)	73.3	73.3	73.3	73.3
F1-score (%)	51.9	90.9	72	74.3

The confusion matrix in Figure 4 shows that the LSTM model yielded good classification accuracy, reaching 73.3%. For the mild class, 15 subjects were correctly classified as mild, and 3 subjects were misclassified as normal. For the moderate class, nine subjects were correctly classified as moderate, and one was misclassified as severe, suggesting some overlap of features between the moderate and severe classes, with no subjects misclassified from other classes. For the severe class, 13 patients were correctly classified as severe, and 7 were incorrectly predicted (2 as moderate and 5 as normal). This shows that there was confusion between the severe and normal classes, and the model had some difficulty distinguishing between these two classes. Finally, for the normal class, seven instances were correctly classified as normal, four instances were incorrectly predicted as moderate, and one was misclassified as severe, demonstrating good recognition capability for the normal class.

Overall, the results for mild and moderate cases showed high true positive rates and low false positive rates, with an acceptable level of misclassified severe subjects incorrectly classified as normal. This can be attributed to similarities in EEG features among these classes. Altogether, having evaluated the proposed model for mild and moderate classification, the results are quite satisfactory.

**Figure 4 bioengineering-12-01225-f004:**
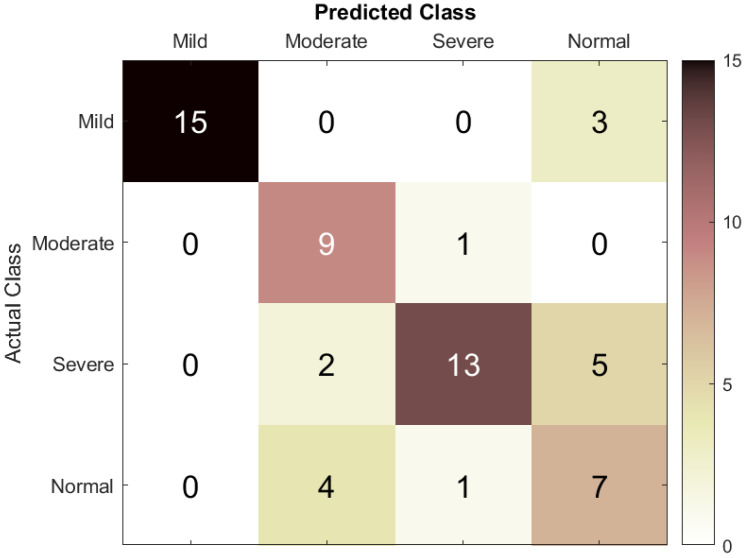
Confusion matrix of the LSTM model used to assess ASD severity from EEG signals.

The ANOVA test showed significant differences in EEG activity among ASD patients with varying severity levels. Specifically, increased severity was associated with a decrease in faster α, β, and γ frequencies and an increase in slower δ and θ frequencies, indicating that each severity level corresponds with unique neurophysiological patterns. Statistical analysis further showed a clear link between ASD severity and changes in EEG frequency powers. The artificial intelligence models applied in this study demonstrated varying levels of effectiveness. The DT classifier achieved 65% accuracy, while the LSTM neural network achieved a higher accuracy of 73.3%. This suggests that these models can effectively differentiate mild, moderate, and severe ASD from typically developing children. This study shows that the combination of statistical analyses, EEG data, and artificial intelligence frameworks is a reliable approach to identifying neurophysical markers of ASD severity. These results could help doctors diagnose and treat ASD earlier by providing a better picture of the changes in the brain and body associated with different severity levels, which would ultimately be better for patients. In conclusion, this study successfully identified EEG patterns associated with ASD severity, demonstrated correlations between EEG frequency changes and ASD severity, and confirmed that deep learning classifiers can effectively distinguish between ASD severity levels, offering potential ways to improve diagnosis and treatment.

## 5. Conclusions

This study aimed to assess the degree of ASD severity based on a statistical analysis of patient data and the use of artificial intelligence frameworks. The observed results emphasize the usefulness of EEG as a diagnostic method for ASD, which can provide valuable data on the electrical manifestations of the disorder depending on the extent of the symptoms. This study also demonstrated the use of statistical analysis to obtain the relative power features for the spectro-spatial profile of ASD severity. The statistical analysis of EEG data, including two-way ANOVA and Pearson’s correlation analyses, identified differences in the EEG patterns of mild ASD, moderate to severe ASD, and TD children. Most significantly, δ and θ activity, aligned with α, β, and γ activity, differed between the regions, manifesting characteristic neurophysiological patterns for different ASD severities.

Among the transfer learning with CNN models we studied in earlier work are AlexNet, ResNet18, GoogLeNet, MobileNetV2, SqueezeNet, ShuffleNet, and EfficientNetb0 [18]. The SqueezeNet model in particular achieved a classification accuracy of 85.5%; when deep features from SqueezeNet were classified using a DT model, a small performance enhancement was observed, yielding an accuracy of 87.8%. In the current work, an RNN with LSTM architecture was applied, which, trained on our ASD EEG dataset, achieved a classification accuracy of 73.3%. Our intention in presenting these comparative results is not to claim superiority but to contextualize our work within the broader field of ASD classification research.

Overall, the investigation showed that the developed LSTM model exhibited better precision and sensitivity, especially for mild and moderate to severe cases of ASD. These results improve the understanding of how deep learning models can infer temporal patterns in EEG data to enhance ASD classification. Therefore, there is an opportunity for the integration of statistical and artificial intelligence frameworks to help develop solutions for the early detection of distinct EEG patterns before the development of clinical symptoms of ASD, potentially allowing earlier intervention. Furthermore, the correct identification of individuals with varying ASD severity levels supports the development of more efficacious treatment plans and, consequently, greater patient satisfaction.

The investigations should be extended to a larger population sample to validate the findings and enhance the external validity of the conclusions, since the present study indicates that dataset limitations may have affected the performance of the model. The existing single-site, single-cohort design will require subsequent confirmation in multiple populations, laboratories, and acquisition devices to reduce possible cultural, procedural, and device biases. Moreover, the LSTM model should be trained on larger datasets to make it more robust and generalizable to ASD groups, which may include multimodal data to identify typical ASD comorbidities such as anxiety or sleep problems. Regarding clinical use, both the proposed spectro-spatial profile of ASD severity and the artificial intelligence systems (such as integrating EEG into ML/DL methods) may be incorporated into real-world ASD diagnostic and treatment workflows to differentiate mild, moderate, and severe ASD patients from typically developing individuals, thereby enhancing clinical decision-making by practitioners. Further research should also focus on optimizing and regularizing DL models and expanding comparisons with a broader range of competitive ML and DL algorithms.

Statistical, machine learning, and deep learning approaches were employed to classify mild, moderate, and severe ASD and to distinguish ASD from typical control data. This helps our understanding of the correlation between multistage ASD and EEG activity by providing a spectro-spatial profile of ASD severity across various brain regions. The obtained level of accuracy makes this study relevant and provides a dependable tool for the early diagnosis of autism. Overall, these findings highlight the potential of the proposed methods to make a valuable contribution to both autism research and clinical practice in the future.

## Figures and Tables

**Table 8 bioengineering-12-01225-t008:** Classification results using the relative power features and DT model for children with mild, moderate, and severe ASD, and typically developing children.

DT	Normal	Mild	Moderate	Severe
Precision (%)	56.4	65.4	70.7	64.7
Sensitivity (%)	44	68	82	66
Specificity (%)	88.7	88	88.7	88
Accuracy (%)	65	65	65	65
F1-score (%)	49.4	66.7	75.9	65.3

## Data Availability

The dataset utilized in this study will be made available upon request to the corresponding author. The data are not publicly available due to stipulations from the institutional ethics review board.

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
