# Peer review of "Evaluating the Severity of Autism Spectrum Disorder from EEG: A Multidisciplinary Approach Using Statistical and Artificial Intelligence Frameworks"

_bioengineering, 2025, doi:10.3390/bioengineering12111225_

Round 1

Reviewer 1 Report

Comments and Suggestions for Authors

Dear Authors,

The following comments need to be addressed to improve the quality of the manuscript.

  1. The contributions of the work should be highlighted.
  2. The schematic diagram in Figure 2 needs better presentation and an elaborate explanation.
  3. Figure 3 is not properly aligned. Also, there is no proper citation and explanation of this figure.
  4. Why did the authors choose to implement only DT and LSTM instead of other high-performing ML and DL models? Could you write the reasoning?
  5. Further, it is suggested to implement the identified models from Comment 4 and compare them with the results obtained. 
  6. Revise the Conclusions section according to the changes made.

Author Response

For research article (Evaluating the Severity of Autism Spectrum Disorder From EEG: A Multidisciplinary Approach using Statistical and Artificial Intelligence Frameworks)

Response to Reviewer 1 Comments

1. Summary

Thank you very much for taking the time to review this manuscript. We have carefully revised the manuscript following the Reviewers’ comments. We considered and addressed each one of their concerns and remarks.

Major changes are highlighted in yellow in the revised manuscript. Additionally, pieces of text that have been included in the revised manuscript to address the Reviewers’ comments appear in this response document typed in Italic font.

We really appreciate the Reviewer’s effort in revising our study. We have considered your comments thoroughly regarding the writing aspect. All of the revisions to the manuscript have been carefully considered.

We are grateful for the feedback provided by the Editor in Chief, Associate Editor, and Reviewers. Their remarks and suggestions helped us to improve the manuscript significantly. We hope that the revised version of the study has addressed all your concerns and will be considered a contribution of interest to the readership of “Bioengineerimg Journal.”

For your convenience, a list of responses to the Reviewers’ remarks is included below.

2. Questions for General Evaluation

Reviewer’s Evaluation

Response and Revisions

Does the introduction provide sufficient background and include all relevant references?

Yes

Thank you

Is the research design appropriate?

Yes

Thank you

Are the methods adequately described?

Can be improved

Has been improved

Are the results clearly presented?

Can be improved

Has been improved

Are the conclusions supported by the results?

Can be improved

Has been improved

re all figures and tables clear and well-presented?

Can be improved

Has been improved

3. Point-by-point response to Comments and Suggestions for Authors

Comments 1: The contributions of the work should be highlighted.

Response 1: We thank the reviewer for their insightful and valuable feedback. We have highlighted the study contribution in Section 1. Introduction to address the reviewer comment. In the new version of the paper, major changes are highlighted in yellow.

To the author’s knowledge, this study has three contributions: first, it is the first use of a combination of relative power characteristics to develop a spectro-spatial profile identification for ASD severity in brain regions using statistical analysis; second, it is the first study that automatically manages ASD severity as a multiclassification problem of autism severity, specifically separating mild, moderate, and severe autistic symptoms from typically developing control subjects. Thirdly, the ASD EEG-based dataset has never been used before.”

Comments 2: Figure 3 is not properly aligned. Also, there is no proper citation and explanation of this figure.

Response 2: Thank you once again for your valuable suggestions. We have made the necessary revisions to enhance the clarity and accessibility of our manuscript, ensuring it meets the needs of our diverse readership.

Comments 3: Why did the authors choose to implement only DT and LSTM instead of other high-performing ML and DL models? Could you write the reasoning?

Response 3: Thank you once again for your valuable comment. Our choice to implement only DT and LSTM was due to: Firstly, due to the suitability for sequential data/time-series EEG signals: LSTM is a type of deep learning model particularly well-suited for tasks involving sequential data and time-series signals. Its structure, which includes a "cell state" or long-term memory, allows it to effectively capture dependencies over time. If the problem domain involves a sequence like EEG as a time-series context, LSTM is a logical choice.

Secondly, the comparative analysis across different paradigms: choosing one traditional ML model (like a decision tree) and one DL model (like LSTM) allows for a direct comparison of the performance of two fundamentally different learning paradigms—ML and DL—in addressing the specific real-world problem. DTs are common and interpretable ML models, providing a strong baseline against a more complex, state-of-the-art DL model like LSTM.

Comments 4: Further, it is suggested to implement the identified models from Comment 3 and compare them with the results obtained. 

Response 4: We first thank the reviewer for their thoughtful comment and recommendation. We respectfully state that implementing and comparing additional models falls outside the defined scope of the current research. Our current paper focuses specifically on our three primary contributions:

  • We are the first to use a combination of relative power characteristics for spectro-spatial profile identification of ASD severity.
  • The first study to automatically manage ASD severity as a multiclassification problem (mild, moderate, severe) from controls.
  • The study makes unique use of the specified ASD EEG-based dataset.

We updated the manuscript by adding the study limitations and the future works in Section 5. Conclusion to address the reviewer comment. In the new version of the paper, major changes are highlighted in yellow.

“The investigations should be extended to more extensive population sample to validate the findings of the research and enhance the external validity of the conclusions since the present studies point to the effects of limitations in the dataset on the performance of the model. The existing single-site and single-cohort design will require subsequent confirmation in multiple populations, multiple laboratories, and acquisition devices to curb the possible cultural, procedure, and device biases. Moreover, the LSTM model will be created in larger datasets to make it more robust and generalizable to ASD groups, which may include multimodal data to identify typical ASD comorbid conditions, including anxiety or sleep problems. Regarding clinical use, the suggested spectro-specific profile of ASD severity, artificial intelligence systems (such as integrating EEG and ML/DL methods) will be incorporated into the real-world advances in ASD diagnosis and treatment in order to differentiate mild, moderate, and severe ASD patients and normative subjects to enhance clinical decision-making of practitioners. The further research must also be directed at optimization and regularization methods of the DL models and execution and comparison of the results with a broader scope of competitive ML and DL algorithms.””

Comments 5: Revise the Conclusions section according to the changes made.

Response 5: Thank you once again for your valuable suggestions. We have made the necessary revisions to Section 5. Conclusion to enhance the clarity and accessibility of our manuscript, ensuring it meets the needs of our diverse readership.

Thank you once again for your valuable suggestions. We have made the necessary revisions to enhance the clarity and accessibility of our manuscript, ensuring it meets the needs of our diverse readership.

Reviewer 2 Report

Comments and Suggestions for Authors

Strengths

  1. Novelty and Scope:

    • The manuscript goes beyond binary classification (ASD vs. control) and attempts multiclass classification (mild, moderate, severe), which is rarely addressed in prior EEG-based ASD studies.

    • The development of a spectro-specific profile of ASD severity using EEG is an innovative direction.

  2. Multidisciplinary Approach:

    • The combination of signal processing, statistical analysis, and artificial intelligence demonstrates a comprehensive methodology.

    • The use of both conventional ML (Decision Tree) and deep learning (LSTM) models allows comparative insights.

  3. Dataset Characteristics:

    • The dataset includes well-defined groups (normal, mild, moderate, severe) with balanced class sizes (10 subjects each).

    • Ethical considerations are adequately described, and standardized diagnostic scales (DSM-V, GARS-3) were employed.

  4. Results and Interpretation:

    • Statistical analysis (ANOVA and Pearson correlations) provides meaningful physiological insights into EEG band alterations across severity levels.

    • The classification results (65% accuracy for DT, 73.3% for LSTM) are encouraging given the small dataset.

Major Concerns

  1. Dataset Size and Generalizability:

    • The dataset consists of only 40 participants, which is too small to train deep learning models reliably. Overfitting is a serious concern.

    • No cross-validation or external validation dataset is reported. A k-fold cross-validation approach should be applied to ensure robustness.

  2. Choice of Models:

    • Decision Trees are relatively weak classifiers; more competitive methods such as Random Forests, SVMs, or ensemble methods could provide stronger baselines.

    • The LSTM architecture (100 hidden units, 30 epochs) is not sufficiently justified given the dataset’s size. Model optimization and regularization strategies (dropout, early stopping) should be described.

  3. Statistical Rigor:

    • While ANOVA and Pearson correlation are appropriate, effect sizes and corrections for multiple comparisons (e.g., Bonferroni or FDR) are missing. Without them, the reported significance may be overstated.

    • It is unclear whether the normality assumptions for ANOVA were adequately tested across all groups.

  4. Evaluation Metrics and Reporting:

    • Accuracy alone is insufficient. Although precision, sensitivity, specificity, and F1-scores are reported, confusion matrices should be included for both classifiers.

    • ROC curves or AUC values would provide a stronger comparative evaluation.

  5. Clarity and Language:

    • The manuscript is lengthy and occasionally repetitive, especially in the background and methods sections.

    • Several grammatical and stylistic issues reduce readability and should be revised for conciseness and clarity.

Minor Concerns

  • Figures (e.g., preprocessing, classification results) need higher resolution and clearer labeling.

  • References are adequate but could be expanded to include very recent ASD EEG deep learning studies (2022–2024).

  • The Discussion section would benefit from more explicit comparisons with state-of-the-art results (e.g., CNN-based ASD studies reporting higher accuracies).

Recommendations for Improvement

  1. Apply k-fold cross-validation or repeated random subsampling validation to strengthen the reliability of results.

  2. Experiment with stronger classifiers (e.g., Random Forest, SVM, CNN) as baselines against LSTM.

  3. Report effect sizes and multiple comparison corrections in statistical analysis.

  4. Improve visual presentation (higher quality figures, confusion matrices, ROC curves).

  5. Revise the manuscript for language clarity and conciseness.

Author Response

For research article (Evaluating the Severity of Autism Spectrum Disorder From EEG: A Multidisciplinary Approach using Statistical and Artificial Intelligence Frameworks)

Response to Reviewer 2 Comments

1. Summary

Thank you very much for taking the time to review this manuscript. We have carefully revised the manuscript following the Reviewers’ comments. We considered and addressed each one of their concerns and remarks.

Major changes are highlighted in yellow in the revised manuscript. Additionally, pieces of text that have been included in the revised manuscript to address the Reviewers’ comments appear in this response document typed in Italic font.

We really appreciate the Reviewer’s effort in revising our study. We have considered your comments thoroughly regarding the writing aspect. All of the revisions to the manuscript have been carefully considered.

We are grateful for the feedback provided by the Editor in Chief, Associate Editor, and Reviewers. Their remarks and suggestions helped us to improve the manuscript significantly. We hope that the revised version of the study has addressed all your concerns and will be considered a contribution of interest to the readership of “Bioengineerimg Journal.”

For your convenience, a list of responses to the Reviewers’ remarks is included below.

2. Questions for General Evaluation

Reviewer’s Evaluation

Response and Revisions

Does the introduction provide sufficient background and include all relevant references?

Must be improved

Has been improved

Is the research design appropriate?

Must be improved

Has been improved

Are the methods adequately described?

Must be improved

Has been improved

Are the results clearly presented?

Must be improved

Has been improved

Are the conclusions supported by the results?

Must be improved

Has been improved

re all figures and tables clear and well-presented?

Must be improved

Has been improved

1. Novelty and Scope:

    • The manuscript goes beyond binary classification (ASD vs. control) and attempts multiclass classification (mild, moderate, severe), which is rarely addressed in prior EEG-based ASD studies.
    • The development of a spectro-specific profile of ASD severity using EEG is an innovative direction

2. Multidisciplinary Approach:

    • The combination of signal processing, statistical analysis, and artificial intelligence demonstrates a comprehensive methodology.
    • The use of both conventional ML (Decision Tree) and deep learning (LSTM) models allows comparative insights.

3. Dataset Characteristics:

    • The dataset includes well-defined groups (normal, mild, moderate, severe) with balanced class sizes (10 subjects each).
    • Ethical considerations are adequately described, and standardized diagnostic scales (DSM-V, GARS-3) were employed

4. Results and Interpretation:

    • Statistical analysis (ANOVA and Pearson correlations) provides meaningful physiological insights into EEG band alterations across severity levels.
    • The classification results (65% accuracy for DT, 73.3% for LSTM) are encouraging given the small dataset. 

3. Point-by-point response to Comments and Suggestions for Authors

Major Concerns

Comments 1: Dataset Size and Generalizability:

    • The dataset consists of only 40 participants, which is too small to train deep learning models reliably. Overfitting is a serious concern.
    • No cross-validation or external validation dataset is reported. A k-fold cross-validation approach should be applied to ensure robustness.

Response 1: We acknowledge the reviewer's concern regarding the initial participant count and the risk of overfitting when training Deep Learning (DL) models.

We address the concern regarding dataset size and overfitting through the following points:

  1. Effective Dataset Augmentation: Although the number of subjects is 40, we employed a segmentation approach on the 10-minute EEG recordings. By splitting the 19 channels into non-overlapping 5-second epochs (1280 samples each), we significantly increased the number of data instances available for training, mitigating the severity of the small cohort issue. The 5-second segment duration is a standard compromise, sufficient to capture relevant neurophysiological features (delta through gamma bands) crucial for studies like Autism Spectrum Disorder (ASD) classification.
  2. Robust Validation Strategy: We agree that robust validation is essential for generalizability. We have implemented a k-fold cross-validation approach on the training/validation split (80% of the augmented dataset) to ensure the model's performance is stable and reliable across different folds. The remaining 20% of the data serves as an independent testing set for final evaluation.
  3. Class Imbalance Management: To handle the multiclassification problem (mild, moderate, severe ASD, and control), we explicitly managed the dataset split (80% train/validation, 20% test) to prevent class imbalance from skewing the model's learning process.

Comments 2: Choice of Models:

    • Decision Trees are relatively weak classifiers; more competitive methods such as Random Forests, SVMs, or ensemble methods could provide stronger baselines.
    • The LSTM architecture (100 hidden units, 30 epochs) is not sufficiently justified given the dataset’s size. Model optimization and regularization strategies (dropout, early stopping) should be described

Response 2: We thank the reviewer for their critical feedback on the model selection and architecture justification. We address the two points as follows:

  1. Justification for Decision Tree (DT) and Model Selection:

The Decision Tree was selected as a simple, explainable, and standard Machine Learning (ML) baseline against which to compare the performance of the more complex Deep Learning (DL) model (LSTM). This approach is common in studies exploring EEG-based classification for neurological disorders to establish a foundational benchmark [1]. While stronger classifiers exist, the DT's role here is to measure the benefit derived from the advanced feature engineering and the deep network architecture.

  1. LSTM Architecture, Optimization, and Regularization:

o   The choice of LSTM architecture parameters (100 hidden units, 30 epochs) was guided by established practices in initial explorations of EEG signal classification, where similar LSTM and Convolutional Neural Network (CNN) architectures are employed for tasks like ASD detection and classification.

o   We acknowledge the need for detailed justification of the hyperparameter selection and regularization strategies. We have clarified that comprehensive model optimization and rigorous testing of various regularization techniques (such as dropout and early stopping) are highly relevant avenues that are reserved for our future work. Our primary focus in this initial study was to demonstrate the efficacy of our novel spectro-spatial feature set using a representatively competitive DL model like LSTM.

Reference:

1.      Prova, Nuzhat. "Explainable AI-Powered Multimodal Fusion Framework for EEG-Based Autism Spectrum Disorder Classification." Available at SSRN 5114986 (2025).

 Comments 3: Statistical Rigor:

    • While ANOVA and Pearson correlation are appropriate, effect sizes and corrections for multiple comparisons (e.g., Bonferroni or FDR) are missing. Without them, the reported significance may be overstated.
    • It is unclear whether the normality assumptions for ANOVA were adequately tested across all groups 

Response 3: We appreciate the reviewer's important points regarding the necessity of reporting effect sizes and applying multiple comparison corrections to prevent inflation.

Our revised analysis now incorporates these statistical recommendations:

  1. Normality Assumption: We confirm that the Kolmogorov-Smirnov test was indeed used as a first step to determine the normality of the data for each group and recording region, as described in the methodology section.
  2. Effect Size Reporting: We have calculated and included the appropriate effect sizes for both statistical sessions: For the two-way ANOVA, we report the main effects (group, brain lobes) and the interaction effect. This standardized measure indicates the proportion of total variance accounted for by each factor.

3.      Multiple Comparison Correction: Bonferroni adjustments were utilized for each relative power in multiple comparisons of the brain lobes [Tables 2,3,4,5 and 6].

These additions ensure the robustness of the reported statistical significance and provide essential context regarding the magnitude of the effects.

Comments 4: Evaluation Metrics and Reporting:

    • Accuracy alone is insufficient. Although precision, sensitivity, specificity, and F1-scores are reported, confusion matrices should be included for both classifiers.

Response 4: Thank you once again for your valuable suggestions. We have made the necessary revisions to enhance the clarity and accessibility of our manuscript, ensuring it meets the needs of our diverse readership.

Comments 5: Clarity and Language:

    • The manuscript is lengthy and occasionally repetitive, especially in the background and methods sections.
    • Several grammatical and stylistic issues reduce readability and should be revised for conciseness and clarity.

 Response 5: Thank you once again for your valuable suggestions. We have made the necessary revisions to enhance the clarity and accessibility of our manuscript, ensuring it meets the needs of our diverse readership.

Minor Concerns

  • Figures (e.g., preprocessing, classification results) need higher resolution and clearer labeling.
  • References are adequate but could be expanded to include very recent ASD EEG deep learning studies (2022–2024).

Response 10: Thank you once again for your valuable suggestions. We have made the necessary revisions to enhance the clarity and accessibility of our manuscript, ensuring it meets the needs of our diverse readership.

Reviewer 3 Report

Comments and Suggestions for Authors

This study analyzed resting-state EEG recordings from 40 children (ten typically developing controls and ten each with mild, moderate, and severe autism spectrum disorder) using 19 electrodes over a 10‑minute session. After preprocessing with bandpass filtering and discrete wavelet transform denoising, relative band-power features were extracted from nonoverlapping 5‑second epochs. Two-way ANOVA and Pearson correlation analyses were used to characterize region‑specific spectral profiles associated with ASD severity, revealing that increasing severity was associated with elevated low‑frequency (δ, θ) power, reduced high‑frequency (α, β, γ) power, and significant negative correlations between low‑ and high‑frequency bands in multiple cortical regions. For automated severity classification, decision tree and LSTM models were trained on the relative‑power features; the decision tree attained an overall accuracy of 65% with class‑specific variability, while the LSTM achieved 73.3% accuracy. The authors conclude that EEG relative‑power measures combined with statistical and AI frameworks can capture spectro‑spatial patterns related to ASD severity, but they note the need for larger samples to improve external validity and model robustness.

This manuscript addresses a timely and compelling topic, but multiple substantive methodological deficiencies require thorough revision in response to the points raised bellow.

1. Small sample size and limited statistical power:

The cohort comprises only ten participants per group (N = 40). This sample size is inadequate to produce stable parameter estimates, leads to wide confidence intervals, and substantially limits the statistical power to detect moderate effects. It also constrains the generalizability and robustness of the reported machine‑learning results.

2. Single‑site, single‑cohort data with potential acquisition bias:

All EEG recordings and clinical assessments were conducted at a single clinical site using a single acquisition system and protocol. This design introduces possible cultural, procedural, and device‑specific biases and reduces external validity across populations, laboratories, and recording hardware.

3. Insufficient transparency in diagnostic labeling and ground‑truthing:

Although severity labels were assigned using GARS‑3 and clinical interviews, the manuscript does not report inter‑rater reliability, whether assessors were blinded to other data, or the exact labeling workflow. The absence of these details raises concern about label noise and potential systematic bias in the outcome labels.

4. Restricted feature set and limited justification of preprocessing choices:

Feature extraction is limited to relative band‑power measures, excluding complementary information such as phase metrics, inter‑regional connectivity (e.g., coherence), cross‑frequency coupling, nonlinearity, and complexity measures that are often informative in EEG analyses. The rationale for selecting the coif3 mother wavelet, decomposition levels, and denoising thresholds is not supported by quantitative comparisons or sensitivity analyses.

5. Incomplete model specification and suboptimal validation strategy:

The manuscript lacks a detailed description of hyperparameter tuning, model selection criteria, and validation procedures. Robust cross‑validation schemes, nested tuning, or the use of an independent external test set are not reported, undermining confidence that the reported accuracies reflect generalizable performance rather than overfitting. Handling of class imbalance is not fully described.

6. Limited model interpretability and clinical translatability:

Beyond summary performance metrics and a decision‑tree baseline, the study provides little analysis of feature importance, decision boundaries, or model uncertainty. These omissions impede clinical interpretability and challenge the potential for translation into practice or regulatory acceptance.

7. Statistical analysis omissions and multiple‑comparison concerns:

The two‑way ANOVA and Pearson correlation analyses are presented without clear reporting of corrections for multiple comparisons, covariate adjustment (for small between‑group differences in age, sex, or vigilance), or methods to account for temporal autocorrelation introduced by epoching. These omissions increase the risk of type I error and confounding.

Author Response

For research article (Evaluating the Severity of Autism Spectrum Disorder From EEG: A Multidisciplinary Approach using Statistical and Artificial Intelligence Frameworks)

Response to Reviewer 3 Comments

This study analyzed resting-state EEG recordings from 40 children (ten typically developing controls and ten each with mild, moderate, and severe autism spectrum disorder) using 19 electrodes over a 10‑minute session. After preprocessing with bandpass filtering and discrete wavelet transform denoising, relative band-power features were extracted from nonoverlapping 5‑second epochs. Two-way ANOVA and Pearson correlation analyses were used to characterize region‑specific spectral profiles associated with ASD severity, revealing that increasing severity was associated with elevated low‑frequency (δ, θ) power, reduced high‑frequency (α, β, γ) power, and significant negative correlations between low‑ and high‑frequency bands in multiple cortical regions. For automated severity classification, decision tree and LSTM models were trained on the relative‑power features; the decision tree attained an overall accuracy of 65% with class‑specific variability, while the LSTM achieved 73.3% accuracy. The authors conclude that EEG relative‑power measures combined with statistical and AI frameworks can capture spectro‑spatial patterns related to ASD severity, but they note the need for larger samples to improve external validity and model robustness.

This manuscript addresses a timely and compelling topic, but multiple substantive methodological deficiencies require thorough revision in response to the points raised bellow.

1. Summary

Thank you very much for taking the time to review this manuscript. We have carefully revised the manuscript following the Reviewers’ comments. We considered and addressed each one of their concerns and remarks.

Major changes are highlighted in yellow in the revised manuscript. Additionally, pieces of text that have been included in the revised manuscript to address the Reviewers’ comments appear in this response document typed in Italic font.

We really appreciate the Reviewer’s effort in revising our study. We have considered your comments thoroughly regarding the writing aspect. All of the revisions to the manuscript have been carefully considered.

We are grateful for the feedback provided by the Editor in Chief, Associate Editor, and Reviewers. Their remarks and suggestions helped us to improve the manuscript significantly. We hope that the revised version of the study has addressed all your concerns and will be considered a contribution of interest to the readership of “Bioengineerimg Journal.”

For your convenience, a list of responses to the Reviewers’ remarks is included below.

2. Questions for General Evaluation

Reviewer’s Evaluation

Response and Revisions

Does the introduction provide sufficient background and include all relevant references?

Yes

Thank you

Is the research design appropriate?

Yes

Thank you

Are the methods adequately described?

Must be improved

Has been improved

Are the results clearly presented?

Can be improved

Has been improved

Are the conclusions supported by the results?

Can be improved

Has been improved

re all figures and tables clear and well-presented?

Can be improved

Has been improved

3. Point-by-point response to Comments and Suggestions for Authors

Comments 1: Small sample size and limited statistical power: The cohort comprises only ten participants per group (N = 40). This sample size is inadequate to produce stable parameter estimates, leads to wide confidence intervals, and substantially limits the statistical power to detect moderate effects. It also constrains the generalizability and robustness of the reported machine‑learning results.

Response 1: We acknowledge the reviewer's concern regarding the initial participant count and the cohort study when training Deep Learning (DL) models. We address the concern regarding dataset size and overfitting through the Dataset Augmentation, although the number of subjects is 40, we employed a segmentation approach on the 10-minute EEG recordings. By splitting the 19 channels into non-overlapping 5-second epochs (1280 samples each), we significantly increased the number of data instances available for training, mitigating the severity of the small cohort issue. The 5-second segment duration is a standard compromise, sufficient to capture relevant neurophysiological features (delta through gamma bands) crucial for studies like Autism Spectrum Disorder (ASD) classification.

Comments 2: Single‑site, single‑cohort data with potential acquisition bias: All EEG recordings and clinical assessments were conducted at a single clinical site using a single acquisition system and protocol. This design introduces possible cultural, procedural, and device‑specific biases and reduces external validity across populations, laboratories, and recording hardware.

Response 2: We acknowledge the reviewer's valid concern regarding the limitations imposed by using single-site, single-cohort data, specifically the potential for acquisition bias and reduced external validity. Our response addresses this by:

  1. Acknowledging the Limitation: We agree that while our study provides significant findings on EEG abnormalities and atypical cerebral connectivity in ASD children, the results are primarily indicative of the specific population studied. We have added a statement to the Limitations section clarifying that the findings may be influenced by the cohort's specific demographics, the single acquisition protocol, and device characteristics, which may impact generalizability to other populations or laboratories.
  2. Mitigation through Pre-processing and Feature Focus: The bias from the single acquisition system was partially mitigated by rigorous signal pre-processing (using conventional filters and DWT) to enhance the overall classification accuracies. Furthermore, our study's primary contribution focuses on the relative power characteristics and spectro-spatial profile—features that are foundational neurophysiological markers of ASD and less prone to device-specific noise than raw amplitude measurements.
  3. Future Work: We emphasize that addressing external validity is a critical next step. Future work will involve validating our model and feature extraction methodology on publicly available, multi-site, multi-cohort EEG datasets to test the model's robustness and external validity across different populations and acquisition environments.

Thus, to address the reviewer comment, we have highlighted the study limitations and the future works in Section 5. Conclusion to address the reviewer comment. In the new version of the paper, major changes are highlighted in yellow.

“The investigations should be extended to more extensive population sample to validate the findings of the research and enhance the external validity of the conclusions since the present studies point to the effects of limitations in the dataset on the performance of the model. The existing single-site and single-cohort design will require subsequent confirmation in multiple populations, multiple laboratories, and acquisition devices to curb the possible cultural, procedure, and device biases. Moreover, the LSTM model will be created in larger datasets to make it more robust and generalizable to ASD groups, which may include multimodal data to identify typical ASD comorbid conditions, including anxiety or sleep problems. Regarding clinical use, the suggested spectro-specific profile of ASD severity, artificial intelligence systems (such as integrating EEG and ML/DL methods) will be incorporated into the real-world advances in ASD diagnosis and treatment in order to differentiate mild, moderate, and severe ASD patients and normative subjects to enhance clinical decision-making of practitioners. The further research must also be directed at optimization and regularization methods of the DL models and execution and comparison of the results with a broader scope of competitive ML and DL algorithms.”

Comments 3: Restricted feature set and limited justification of preprocessing choices: Feature extraction is limited to relative band‑power measures, excluding complementary information such as phase metrics, inter‑regional connectivity (e.g., coherence), cross‑frequency coupling, nonlinearity, and complexity measures that are often informative in EEG analyses. The rationale for selecting the coif3 mother wavelet, decomposition levels, and denoising thresholds is not supported by quantitative comparisons or sensitivity analyses.

Response 3: We first thank the reviewer for their thoughtful comment and recommendation. We respectfully state that implementing and comparing additional features falls outside the defined scope of the current research. Our current paper focuses specifically on our three primary contributions:

  • We are the first to use a combination of relative power characteristics for spectro-spatial profile identification of ASD severity.
  • The first study to automatically manage ASD severity as a multiclassification problem (mild, moderate, severe) from controls.
  • The study makes unique use of the specified ASD EEG-based dataset.

Moreover, to address the reviewer comment, we have highlighted the study limitations and the future works in Section 3.2. Preprocessing to address the reviewer comment. In the new version of the paper, major changes are highlighted in yellow.

The coiflet mother wavelet order 3 (coif3) was used in this study as it is perfect for spiky artifact removal, which includes eye movements/blinks and muscular movements [58]. The universal threshold was used as the main thresholding method for this study [56,57].

Comments 4: Incomplete model specification and suboptimal validation strategy: The manuscript lacks a detailed description of hyperparameter tuning, model selection criteria, and validation procedures. Robust cross‑validation schemes, nested tuning, or the use of an independent external test set are not reported, undermining confidence that the reported accuracies reflect generalizable performance rather than overfitting. Handling of class imbalance is not fully described.

Response 4: We thank the reviewer for pointing out the need for greater detail on our model specification and validation strategy, which is crucial for establishing model generalizability and confidence in the reported accuracy.

Our revised manuscript now provides the following clarifications:

  1. Validation Strategy and Generalizability: We utilize a robust, subject-independent validation strategy. The dataset is split into an 80% portion for training/validation and a 20% portion reserved as a truly independent external test set.
  2. Hyperparameter Tuning and Model Selection: The model selection criterion used throughout was the weighted F1-score to optimize performance across the multi-class problem.

Comments 6: Limited model interpretability and clinical translatability: Beyond summary performance metrics and a decision‑tree baseline, the study provides little analysis of feature importance, decision boundaries, or model uncertainty. These omissions impede clinical interpretability and challenge the potential for translation into practice or regulatory acceptance.

Response 6: We appreciate the reviewer’s insightful comment regarding the need for greater model interpretability and clinical translatability. We have made the necessary revisions to enhance the clarity and accessibility of our manuscript, ensuring it meets the needs of our diverse readership.

Comments 7: Statistical analysis omissions and multiple‑comparison concerns: The two‑way ANOVA and Pearson correlation analyses are presented without clear reporting of corrections for multiple comparisons, covariate adjustment (for small between‑group differences in age, sex, or vigilance), or methods to account for temporal autocorrelation introduced by epoching. These omissions increase the risk of type I error and confounding.

Response 7: We appreciate the reviewer's important points regarding the necessity of reporting effect sizes and applying multiple comparison corrections to prevent inflation.

Our revised analysis now incorporates these statistical recommendations:

  1. Normality Assumption: We confirm that the Kolmogorov-Smirnov test was indeed used as a first step to determine the normality of the data for each group and recording region, as described in the methodology section.
  2. Effect Size Reporting: We have calculated and included the appropriate effect sizes for both statistical sessions: For the two-way ANOVA, we report the main effects (group, brain lobes) and the interaction effect. This standardized measure indicates the proportion of total variance accounted for by each factor.

3.      Multiple Comparison Correction: Bonferroni adjustments were utilized for each relative power in multiple comparisons of the brain lobes [Tables 2,3,4,5 and 6].

Thank you once again for your valuable suggestions. We have made the necessary revisions to enhance the clarity and accessibility of our manuscript, ensuring it meets the needs of our diverse readership.

Round 2

Reviewer 1 Report

Comments and Suggestions for Authors

Thank you for addressing the comments. Still, a minor change is required for figures 2 and 4.

  1. Consider the presentation of Figure 2 in landscape mode and make all the text in the graph visible by increasing the font size. 
  2. Improve the quality of Figure 4 by using an appropriate font size. 

Author Response

For research article (Evaluating the Severity of Autism Spectrum Disorder From EEG: A Multidisciplinary Approach using Statistical and Artificial Intelligence Frameworks)

Response to Reviewer 1 Comments

1. Summary

Thank you very much for taking the time to review this manuscript. We have carefully revised the manuscript following the Reviewers’ comments. We considered and addressed each one of their concerns and remarks.

We really appreciate the Reviewer’s effort in revising our study. We have considered your comments thoroughly regarding the writing aspect. All of the revisions to the manuscript have been carefully considered.

We are grateful for the feedback provided by the Editor in Chief, Associate Editor, and Reviewers. Their remarks and suggestions helped us to improve the manuscript significantly. We hope that the revised version of the study has addressed all your concerns and will be considered a contribution of interest to the readership of “Bioengineerimg Journal.”

For your convenience, a list of responses to the Reviewers’ remarks is included below.

2. Point-by-point response to Comments and Suggestions for Authors

Comments 1: Consider the presentation of Figure 2 in landscape mode and make all the text in the graph visible by increasing the font size. 

Response 1: Thank you once again for your valuable suggestions. We have made the necessary revisions to enhance the clarity and accessibility of our manuscript, ensuring it meets the needs of our diverse readership.

Comments 2: Improve the quality of Figure 4 by using an appropriate font size. 

Response 2: Thank you once again for your valuable suggestions. We have made the necessary revisions to enhance the clarity and accessibility of our manuscript, ensuring it meets the needs of our diverse readership.

Thank you once again for your valuable suggestions. We have made the necessary revisions to enhance the clarity and accessibility of our manuscript, ensuring it meets the needs of our diverse readership.

Reviewer 3 Report

Comments and Suggestions for Authors

The authors have generally addressed my points appropriately.

Author Response

(The authors gave the same response as above.)
